# Analysis of Electromagnetic Shielding Properties of a Material Developed Based on Silver-Coated Copper Core-Shell Spraying

**DOI:** 10.3390/ma15155448

**Published:** 2022-08-08

**Authors:** Yu-Jae Jeon, Jong-Hwan Yun, Min-Soo Kang

**Affiliations:** 1Department of Medical Rehabilitation Science, Yeoju Institute of Technology, Yeoju 12652, Korea; 2Regional Innovation Platform Project, Kongju National University, Cheonan 31080, Korea; 3Division of Smart Automotive Engineering, Sun Moon University, Asan 31460, Korea

**Keywords:** electromagnetic shielding, shielding material, silver-coated copper, core shell, electromagnetic interference (EMI)

## Abstract

This study proposes an electromagnetic shielding material sprayed with silver-coated copper powder (core-shell powder). The shielding properties of the material are analyzed in details section. Cross-sectional observation and sheet resistance measurement were used to determine the thickness and electrical conductivity of the electromagnetic shielding layer, which was generated by spray-coating; this aided in confirming the uniformity of the coating film. The results indicate that the electromagnetic interference shielding effectiveness increases when the silver-coated copper paste (core-shell paste) is used as the coating material rather than the conventional aluminum base. The proposed material can be used in various frequency ranges owing to the excellent shielding effectiveness of the core-shell paste used in this study. Further investigations on the optimized spray-coating type of electromagnetic shielding material are required based on the composition of the core-shell paste and the thickness of the coating film.

## 1. Introduction

The space around us is filled with electromagnetic waves of various frequencies. Essential technologies, such as radio navigation, mobile and satellite communications, television and radio broadcasting, and Wi-Fi operate in the range of tens of kHz to hundreds of GHz. To accurately operate high-precision electronic devices and lifesaving systems, the devices should be protected from parasitic radio emissions in addition to protecting their stored information. Therefore, different materials, designs, and technological solutions have been explored that provide radio shielding in various frequency ranges [1]. The recent advent of the 5G/6G era has resulted in electromagnetic interference (EMI) emerging as a severe problem because of the large-scale utilization of electronic devices in the telecommunication and electrical industries [2,3,4]. As the rapid development of 5G communication, high-speed wireless communication, and the supply expansion of portable wearable electronic devices based on the interfering communication signals affect human health adversely, electromagnetic radiation is considered a global concern [5,6]. Therefore, the electromagnetic shield has been increasingly investigated in recent years. Shielding materials are crucial for protecting electronic equipment from EMI and preventing the radiation sources from emitting electromagnetic waves. EMI shielding refers to the absorption or reflection of electromagnetic waves using effective shielding materials that are composed of either conductive or magnetic materials [7,8]. Although metals have been widely used as EMI shielding materials owing to their excellent shielding performance, they exhibit certain drawbacks, such as low corrosion resistance and poor mechanical flexibility [9,10,11]. Nevertheless, electrically conductive metals exhibit excellent EMI shielding performance and have been widely used in EMI shielding applications [12,13,14]. However, unlike polymers and ceramics, the relatively higher densities of metals result in a tradeoff between shield weight and shielding performance. Additionally, as metals are susceptible to corrosion and environmental degradation, structural integrity and EMI shielding performance are known to deteriorate. These considerations have led to the development of lighter polymeric EMI shielding materials [15]. As most polymers exhibit extremely low electrical conductivity, EMI shielding performance can be improved via two approaches, namely coating or blending with conductive materials to generate composites [16]. Typically, conductive fillers, such as carbon nanofibers [17,18], carbon nanotubes [19,20,21,22], graphene sheets [23,24,25,26], and metal wires and particles [27,28,29] are used to synthesize polymer-based composites. Recently, coating of particles and the design of coating processes have been comprehensively investigated in several industrial fields because coating improves and changes the surface properties or functionalities of particles with respect to catalytic activity, hardness, permeability, adhesion, and conductivity [30]. Based on the aforementioned studies, researchers have succeeded in designing a useful novel hybrid composite additive, referred to as the core-shell (core@shell or core/shell). The heterogeneous core-shell particles are composed of two or more materials, including a metal, an element, or biomolecules. One material serves as a core in the center, whereas the other material or substance is coated on the surface of the center core. Fundamentally, the core-shell particle is a type of biphasic nanomaterial, wherein an inner core and outer shell are composed of different components [31]. The coating material of the core-shell protects the core and inhibits its oxidation [32]. Copper particles of 20 wt% of silver loading were stable under air and 95% of copper remained metallic copper even after 1 month of exposure to air. This enhanced air-stability contributed to the enhanced electrical property of conductive film obtained from the coated particles [33]. Thus, coating the core-shell of the conductive metal on the surface facilitates electromagnetic shielding. Therefore, to investigate the possibility of improving the electromagnetic shielding performance by coating the existing printed circuit boards and other electronic components with a conductive paste, this study focused on preparing a specimen by spraying a conductive paste. Additionally, a comparative analysis of electromagnetic shielding properties was performed to validate the feasibility of the proposed material.

## 2. Experimental

### 2.1. Materials Specimen Preparation

Silver-coated copper powder (core-shell powder) uses low-cost copper powder as the filler material and silver coating to suppress oxidation, which is the drawback of copper powder. Performing the silver coating via a wet process is advantageous with respect to the manufacturing cost. In this scenario, the degree of coverage, uniformity, thickness, microstructure, and density of the silver coating layer on the copper surface directly affect the oxidation characteristics of copper in the silver coating [34,35,36]. This conductive additive can improve the electromagnetic shielding properties. Furthermore, appropriately blending this powder-type nanoparticle with polymer facilitates its easy coating through the injection or spraying process; alternatively, the specimen can be directly fabricated via an injection process.

Epoxy oligomer, monomer, and a curing agent were used as matrix materials to prepare the core-shell paste from the core-shell powder. The metal powder solid content was 86 wt% in the core-shell paste. The paste mixing and three-roll mill process were performed for dispersion, wherein the polymerization was conducted for 2 h at 70 °C at a stirring rate of 200 rpm. The viscosity was adjusted by adding a solvent for a smooth spraying process. Table 1 lists the properties of the core-shell powder used in this study.

The prepared core-shell paste was sprayed on an aluminum substrate using a spray gun. Figure 1 depicts the conditions of the spraying process, with an injection pressure of 3 Bar, injection time of 3 s, and injection height of 450 mm. Subsequently, a specimen with a coating thickness of 30 µm was produced through a drying process. Through the spraying process, there were a total of 10 specimens manufactured, and the electromagnetic shielding characteristics were compared and analyzed through these specimens.

### 2.2. Measurement of Shielding Effectiveness (SE)

There are many EMI shielding mechanisms, but EMI shielding is largely caused by reflection, absorption, multi-reflection, or internal reflection mechanisms. Generally, electromagnetic reflection is typical. In metallic materials, when electromagnetic waves traveling in a certain direction pass through the shielding material, electromagnetic waves are reflected due to the difference in impedance between the medium through which electromagnetic waves pass on the surface of the material. In addition, there is an electromagnetic absorption mechanism in which electromagnetic waves such as conductive loss, dielectric loss, and magnetic loss are absorbed, converted into thermal energy, and then lost in the electromagnetic absorption. Most of these electromagnetic wave absorption mechanisms occur in carbon-based and magnetic-based materials. In addition, when the material is non-uniform or thick, there is an electromagnetic multi-reflection or electromagnetic internal reflection mechanism in which electromagnetic waves are transmitted in a completely different direction along the material without transmitting the material by electromagnetic scattering. To improve the shielding performance according to such a mechanism, a material with excellent permittivity, permeability, or high conductivity should be selected. In addition, the EMI shielding method using reflection may be determined by the material and the frequency of the incident but may be largely divided into a single-layer metallic shield and a multi-layer metallic shield. The single-layer metallic shield shielding method refers to the aforementioned reflection loss, abortion loss, multi-reflection loss, and internal reflection loss. The multiple shielding effect may increase the shielding effect by adjusting the impedance and thickness of the shielding plates. In conclusion, to improve the EMI shielding effect, a material with excellent permittivity, permeability, or high conductivity should be used. To measure SE, the geometry of the specimen was prepared as depicted in Figure 2 according to the standard ASTM D 4935. Figure 3 illustrates the electromagnetic shielding measurement equipment that was used to analyze the SE of the proposed material; the measurement was obtained in the range of 1.5 to 10 GHz.

Equation (1) indicates the calculation of SE, and the electromagnetic shielding measurements in this study were obtained using the same equation. The reference specimen without the electromagnetic shielding material was used in the electromagnetic shielding measurement to obtain a reference value. Additionally, the SE of the specimen coated with the electromagnetic shielding material was measured for comparative analysis.
(1)$$\mathrm{SE}=20log\frac{V_{2}}{V_{1}}\left(\mathrm{dB}\right)$$

*V*_1_: Received voltage when the shielding material is present. *V*_2_: Received voltage in the absence of the shielding material.

## 3. Results

In the results of this study, one specimen most like the average of the electromagnetic shielding rate measurement results was selected and analyzed among the specimens. Figure 4 depicts the results of the cross-sectional observation that was performed to determine the injection uniformity of the core-shell paste used in this study. The specimen coated with the electromagnetic shielding material was cut, and the cross-section was observed using a scanning electron microscope (SEM). The cross-sectional observation confirmed that the core-shell particles were homogeneously dispersed in the matrix of the core-shell paste and were coated with a thickness of 30 μm.

The uniform coating thickness was considered an indicator of excellent electrical conductivity of the material, owing to the contact of conductive core-shell particles. In general, materials with adequate electrical conductivity result in excellent electromagnetic shielding performance. Therefore, the cross-section indicated that the core-shell paste used in this study exhibits superior electromagnetic shielding performance.

Figure 5 depicts the component analysis performed after injecting the core-shell paste; the presence of impurities was determined in the core-shell particles using energy-dispersive X-ray spectroscopy (EDX). The peaks of Cu and Ag indicated that Cu and Ag were 87.13 and 12.87 wt%, respectively, verifying that no oxidation or impurity formation occurred during the coating process. In EDX, a specimen excited by an energy source releases a portion of the absorbed energy by emitting electrons. The higher-energy outer-shell electrons occupy the positions of the emitted electrons, releasing the energy difference as X-rays with a characteristic spectrum based on the atom of origin. Consequently, the composition of a particular specimen excited by the energy source can be analyzed. The peak position in the spectrum identifies the element, and the intensity of the signal indicates the concentration of the element; this can determine the basic properties of the dispersed material.

Furthermore, the sheet resistance of the coated specimen was measured to evaluate the shielding properties of the core-shell paste. The sheet resistance (Ω/□) of the specimen was determined using a four-point probe (CMT-100S/J) by measuring the current and voltage using four probes; the value is expressed in terms of surface resistance of ohm/sq. Table 2 summarizes the numerical values measured using the correction factor. Based on the measurement results, the average of the sheet resistances at five locations was calculated as 0.1336 Ω/□ ± 5%. The resistance was measured to be nearly equal regardless of the surface location, confirming that the coating film generated by the spraying process was uniformly thick.

The electromagnetic interference SE is expressed in decibel (dB) and can be implemented differently in the range of 0–70 dB for a polymer composite material. Typically, SE of 30 dB or more is considered practical. Particularly, an excellent SE of approximately 100 dB can be obtained in the case of metals in the absence of a seam or hole. The technical average value of SE of the general commercial electromagnetic shielding materials used in peripheral electronic devices, such as smartphones and computers, is in the range of 30 to 60 dB [37,38].

The test in this study was conducted according to ASTM D 4935 in the range of 1.5 to 10 GHz to measure the electromagnetic interference SE. The SE of the uncoated aluminum specimen was measured as a reference for the comparative analysis of the shielding performance of the core-shell paste. The result indicated that the SE varied as a function of frequency; however, the average measurement was determined to be 64.57 dB.

The electromagnetic shielding performance of the specimen was measured with a 30-μm-coating of the core-shell paste on the aluminum baseplate. The results indicate that SE increases to an average of 77.27 dB as shown in Figure 6. As SE is expressed in dB, the shielding rates were determined to be 90, 99, 99.9, 99.99, and 99.999% for 20, 40, 60, 80, and 100 dB, respectively. As a value of 40 dB or more is close to 100%, the SE of the core-shell paste used in this study can be considered excellent. This cause may be due to characteristics of a core-shell structure made of a material with high electrical conductivity. The electrical conductivity of copper is known to be 5.98 × 10^7^ and that of silver is known to be 6.30 × 10^7^. This high electrical conductivity of copper and silver makes it easy to increase EM reflection, so it can be judged that the performance of EM shielding has improved. Furthermore, the core-shell structure is a structure in which a copper metal with a high degree of oxidation is coated on the inside and silver with a low degree of oxidation on the outside, thereby protecting the oxidation of the copper metal. Therefore, since it is smooth to maintain electrical conductivity for a long period of time, it may be used as a material capable of improving electromagnetic wave shielding characteristics. It is also more advantageous with respect to price than Ag powder, depending on the characteristics of cheap copper. According to these characteristics, if the process is combined with a simple spraying process, the composite material for an electromagnetic wave shielding coating film could have a great effect. Additionally, core-shell nanoparticles have garnered special scientific interest as they exhibit certain unique properties owing to their design, core geometry, shell, and the combination of core and shell materials. Particularly, the core-shell nanostructures are effective because of their optimal morphology, adjustable pore size (nanoparticles with porous shell), and free space between the core and shell, which imparts more stability under harsh environments [39]. Therefore, they have been used in multiple fields, including medicine, engineering, electronics, and material science [40,41,42,43,44,45,46,47,48,49,50,51]. Although core-shell nanostructures are small, they are dominant entities with high thermal and chemical stabilities, low toxicity, high solubility, and high permeability to specific target cells [39]. Based on previously reported results, the properties of nanostructures vary depending on the core-shell particle size, shape, and material. In the future, a comparative analysis of SE should be performed according to the coating thickness of the core-shell paste and the wt% of the core-shell powder, and further studies on the oxidation resistance of the core-shell should be conducted.

## 4. Conclusions

This study proposes a spray-type core-shell paste composite material blended with core-shell powder. The specimen was prepared by spraying, and the sheet resistance of the specimen was measured to evaluate the precision and reproducibility of the spraying process. The thickness of the sprayed coating was evaluated by cross-sectional observations. Additionally, the precision of specimen preparation was determined. An electromagnetic shielding measurement test was conducted to measure SE according to ASTM D 4935. The SE was determined using the difference between the numerical values of the electromagnetic wave transmittances measured from the coated and reference specimens. The study findings can be summarized as follows. The thickness of the electromagnetic interference shielding film coated via the spraying process, which was suitable for the core-shell paste characteristics, was 30 μm. The sheet resistance of the coated specimen measured to ensure reproducibility of the specimen indicated that the coating was homogenous with 0.1336 Ω/□ ± 5%. Thus, the possibility of a precise coating was confirmed. Moreover, the measured electromagnetic interference SE of the base aluminum plate was 64.57 dB, whereas that of the specimen coated with the core-shell paste was 77.27 dB. This indicates an improvement in SE by 120% compared to that of the reference specimen. In the future, the optimized composition that can increase SE in various bands should be determined by performing experiments with respect to the coating thickness of the core-shell paste, wt% of the core-shell powder, core-shell particle size, shape, and material. In the future, the optimized composition that can increase SE in various bands should be determined by performing experiments with respect to the coating thickness of the core-shell paste, wt% of the core-shell powder, core-shell particle size, shape, and material.

## Figures and Tables

**Figure 1 materials-15-05448-f001:**
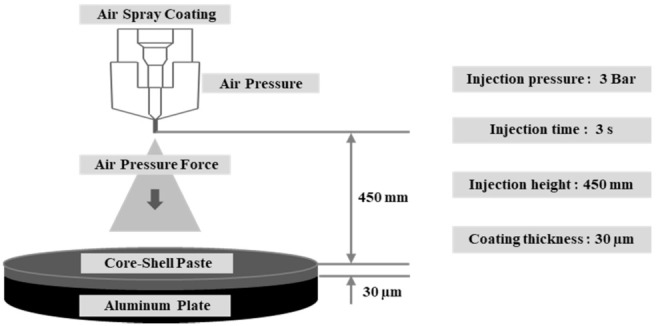
Schematic of the core-shell paste spraying process.

**Figure 2 materials-15-05448-f002:**
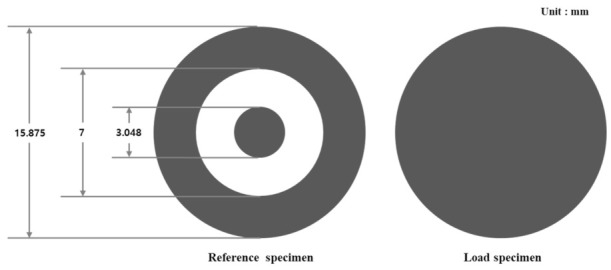
Geometry of the specimen for electromagnetic shielding measurement.

**Figure 3 materials-15-05448-f003:**
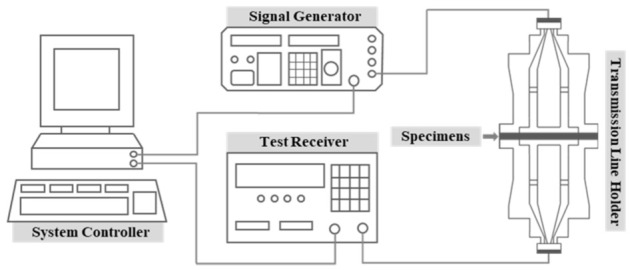
Schematic of the electromagnetic shielding measurement test.

**Figure 4 materials-15-05448-f004:**
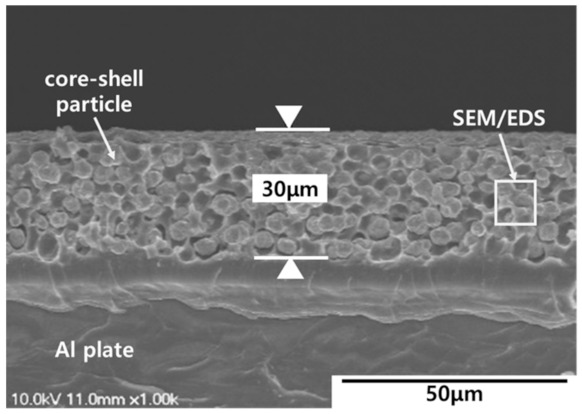
Cross-sectional image of the core-shell paste coating.

**Figure 5 materials-15-05448-f005:**
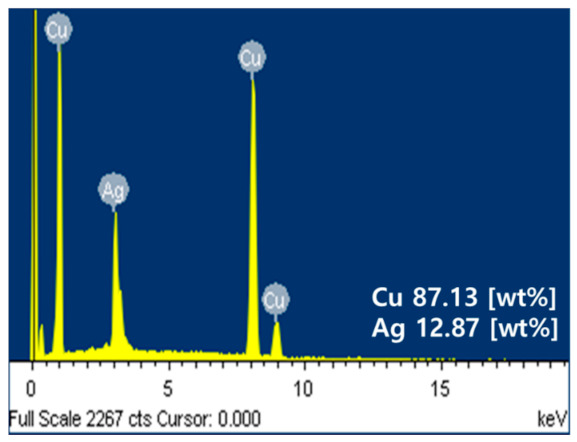
Component analysis of the core-shell.

**Figure 6 materials-15-05448-f006:**
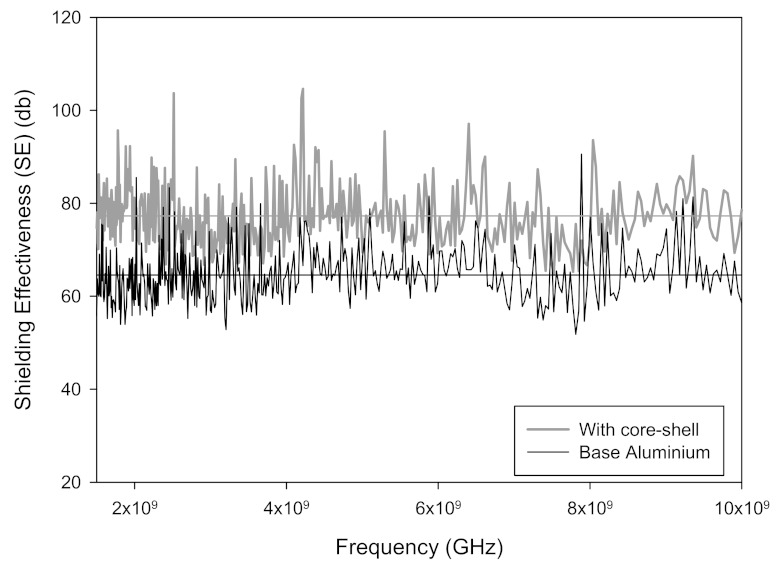
Shielding effectiveness as a function of frequency.

**Table 1 materials-15-05448-t001:** Properties of the core-shell powder.

Features	Contents
State	Solid
Appearance	Powder
Particle size	2.0 µm, 4.0–4.4 µm, 6.0 µm
Particle shape	Spherical
Specific surface area	0.40 m^2^/g
Bulk density	4.2 g/m^3^
Purity	99.9%
Alternate Name	Silver-coated copper powder

**Table 2 materials-15-05448-t002:** Measurement of sheet resistance using a four-point probe.

Location	Sheet Resistance (Ω/□)
1	0.1309
2	0.1410
3	0.1254
4	0.1321
5	0.1387

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
