# Peer review of "Analysis of Electromagnetic Shielding Properties of a Material Developed Based on Silver-Coated Copper Core-Shell Spraying"

_materials, 2022, doi:10.3390/ma15155448_

Round 1
Reviewer 1 Report
This paper proposes an electromagnetic shielding material sprayed with silver-coated copper powder (core-shell powder). This is quite interesting and good work. I congratulate the authors on this state-of-the-art research.
Author Response
Thank you for your Review Report. We will be continue to research and announce the results in materials journal.
Additionally, if you have any other discussions or opinions, please contact us.
Once again, thank you.
Reviewer 2 Report
The authors have presented an investigation proposing an electromagnetic shielding material sprayed with silver-coated copper powder (core-shell powder). The paper is well written and contains backings from experimental results. However, for the manuscript to be qualified to be accepted for publication in the Materials Journal, I would like to suggest the following corrections:
1. The following sentence (Line 66-67) below needs a reference “Coating the core-shell of the conductive metal on the surface facilitates electromagnetic shielding”
2. Line 73: The materials used need to be highlighted in a dedicated subsection. E.g., 2.1 Materials
3. The paragraph containing lines 75-84 is supposed to come under introduction part. Please shift it
4. The two sub-sections in section 2 are numbered 2.1. Please correct
5. Additional analytical techniques capable of confirming the successful preparation of the composite material are needed.
6. I suggest the itemization in the conclusion to be removed and changed to a paragraph

Author Response
Thank you.
Your good review comments have helped us to revise our article
The results are as follows.
1. The following sentence (Line 66-67) below needs a reference “Coating the core-shell of the conductive metal on the surface facilitates electromagnetic shielding”
- Additional explanatory sentence on Core-shell were added along with reference papers as below in Line 65-69.
“Copper particles of 20 wt.% of silver loading were stable under air and 95% of copper remained as metallic copper even after 1 month of exposure to air. This enhanced air-stability contributed to the enhanced electrical property of conductive film obtained from the coated particles[33].”
<Reference>
[33] “D.S. Jung, H.M. Lee, Y.C. Kang, S.B. Park (2011), Air-stable silver-coated copper particles of sub-micrometer size. Journal of Colloid and Interface Science, https://doi/10.1016/j.jcis.2011.08.033” Thus, coating the core-shell of the conductive metal on the surface facilitates electromagnetic shielding.
2. Line 73: The materials used need to be highlighted in a dedicated subsection. E.g., 2.1 Materials
- Subheadings have been modified as below.
“2.1 Materials specimen preparation”
3. The paragraph containing lines 75-84 is supposed to come under introduction part. Please shift it
- That explains the characteristics of the core shell and explains the characteristics of the nanoparticles produced for the spray process. If transferred to the introduction, an overlapping description of why the core shell was used in the manufacturing process may be required. I respect the reviewer's opinion, but if you also consider not moving the paragraph, I would appreciate it.
4. The two sub-sections in section 2 are numbered 2.1. Please correct
- We revised the subheading as below.
"2.2 Measurement of Shielding Effectiveness (SE)"
5. Additional analytical techniques capable of confirming the successful preparation of the composite material are needed.
- In fact, we agree with the reviewer's opinion that more evaluation test should be performed according to the manufacture of core-shell composite materials. However, at the time of writing this paper, no special composite material evaluation technology was used Because it's in the early stages of research. In future research papers, we will analyze and present the evaluation of the adhesive force of the film coated with the composite material through the spraying process, the viscosity of the composite material, and the DSC characteristics.
6. I suggest the itemization in the conclusion to be removed and changed to a paragraph
- As shown below, we revised the paragraph of the conclusion.
“The thickness of the electromagnetic interference shielding film coated via the spraying process, which was suitable for the core-shell paste characteristics, was 30 μm. The sheet resistance of the coated specimen measured to ensure reproducibility of the specimen indicated that the coating was homogenous with 0.1336 Ω/□±5%. Thus, the possibility of a precise coating was confirmed. Moreover, the measured electromagnetic interference SE of the base aluminum plate was 64.57 dB, whereas that of the specimen coated with the core-shell paste was 77.27 dB. This indicates an improvement in SE by 120% compared to that of the reference specimen. In the future, the optimized composition that can increase SE in various bands should be determined by performing experiments in terms of the coating thickness of the core-shell paste, wt% of the core-shell powder, core-shell particle size, shape, and material.”
Reviewer 3 Report
This study proposes an electromagnetic shielding material sprayed with silver-coated copper powder. Cross-sectional observation and sheet resistance measurement were used to determine the thickness and electrical conductivity of the electromagnetic shielding layer, which was generated by spray coating.
1 In this paper, only one spraying experiment has been done and only one sample has been analyzed. It is suggested to supplement the experiment to study the influence of different parameters on the electromagnetic shielding performance of the coating.
2 In the Introduction, it is said that the core-shell coating material protects the core and inhibits its oxidation (Line 65). However, this study only found that the coating can improve the shielding effectiveness of electromagnetic interference, and did not study the oxidation resistance of the coating. It is suggested to supplement the test and analysis of oxidation resistance of coating.
3 Why the coating in this study can improve the shielding effectiveness of electromagnetic interference needs to be analyzed in more detail.
Author Response
Thank you
Your good comments helped us a lot to revise our paper.
The following are the revisions to the article.
1. In this paper, only one spraying experiment has been done and only one sample has been analyzed. It is suggested to supplement the experiment to study the influence of different parameters on the electromagnetic shielding performance of the coating.
- We didn't just perform a single spray experiment. Ten samples were measured, analyzed, and samples closest to the average of the measured shielding rate were selected, described, and analyzed. If the entire graph for the sample is represented, the noise of the graph will increase. So, we marked only one sample.
- We added the contents of the number of specimens manufactured below to the third paragraph of "2.1 Specimen Preparation".
"Through the spraying process, there were a total of 10 specimens manufactured, and the electromagnetic shielding characteristics were compared and analyzed through these specimens."
- In addition, the following was added to the first paragraph of "Results".
"In the results of this study, one specimen most similar to the average of the electromagnetic shielding rate measurement results was selected and analyzed among the specimens."
2. In the Introduction, it is said that the core-shell coating material protects the core and inhibits its oxidation (Line 65). However, this study only found that the coating can improve the shielding effectiveness of electromagnetic interference, and did not study the oxidation resistance of the coating. It is suggested to supplement the test and analysis of oxidation resistance of coating.
- Research on oxidation resistance will be published in addition to the paper later. Currently, we are conducting a study on how much core-shell coating material affects shielding and how appropriate the blending ratio is. In the future, we will check the electromagnetic shielding performance in the core-shell coating material of the optimized composition and submit an additional paper that tested the oxidation resistance of the material. If research results that improve oxidation resistance are derived from future research results, it is judged that management and supervision costs in the core-shell spraying composite materials manufacturing process for EMI shielding may be reduced.
3. Why the coating in this study can improve the shielding effectiveness of electromagnetic interference needs to be analyzed in more detail.
- The following was added at the end of the paragraph "Results".
"This cause may be due to characteristics of a core-shell structure made of a material having high electrical conductivity. The electrical conductivity of copper is known to be 5.98×107 and that of silver is known to be 6.30×107. This high electrical conductivity of copper and silver makes it easy to increase EM reflection, so it can be judged that the performance of EM shielding has improved. Furthermore, the core-shell structure is a structure in which a copper metal with a high degree of oxidation is coated on the in-side and silver with a low degree of oxidation on the outside, thereby protecting the oxidation of the copper metal. Therefore, since it is smooth to maintain electrical conductivity for a long period of time, it may be used as a material capable of improving electromagnetic wave shielding characteristics. It is also more advantageous in terms of price than Ag powder depending on the characteristics of cheap copper. According to these characteristics, if the process is combined with a simple spraying process, the composite material for an electromagnetic wave shielding coating film could have a great effect. In the future, a comparative analysis of SE should be performed according to the coating thickness of the core-shell paste and the wt% of the core-shell powder. and further studies on the oxidation resistance of the core-shell should be conducted."

Round 2
Reviewer 2 Report
The authors have tried in responding to the issues raised. I believe the paper is now good enough to be published in the journal of Materials-MDPI
Reviewer 3 Report
I'm very glad that the author of the paper has revised and supplemented it according to the review comments